# In Vitro Approaches to Determine the Potential Carcinogenic Risk of Environmental Pollutants

**DOI:** 10.3390/ijms24097851

**Published:** 2023-04-25

**Authors:** Irene Barguilla, Veronique Maguer-Satta, Boris Guyot, Susana Pastor, Ricard Marcos, Alba Hernández

**Affiliations:** 1Group of Mutagenesis, Department of Genetics and Microbiology, Faculty of Biosciences, Universitat Autònoma de Barcelona, 08193 Cerdanyola del Vallès, Spain; 2CNRS UMR5286, Centre de Recherche en Cancérologie de Lyon, 69008 Lyon, France

**Keywords:** in vitro cell transformation, oncogenic phenotype, cancer hallmarks, long-term exposure

## Abstract

One important environmental/health challenge is to determine, in a feasible way, the potential carcinogenic risk associated with environmental agents/exposures. Since a significant proportion of tumors have an environmental origin, detecting the potential carcinogenic risk of environmental agents is mandatory, as regulated by national and international agencies. The challenge mainly implies finding a way of how to overcome the inefficiencies of long-term trials with rodents when thousands of agents/exposures need to be tested. To such an end, the use of in vitro cell transformation assays (CTAs) was proposed, but the existing prevalidated CTAs do not cover the complexity associated with carcinogenesis processes and present serious limitations. To overcome such limitations, we propose to use a battery of assays covering most of the hallmarks of the carcinogenesis process. For the first time, we grouped such assays as early, intermediate, or advanced biomarkers which allow for the identification of the cells in the initiation, promotion or aggressive stages of tumorigenesis. Our proposal, as a novelty, points out that using a battery containing assays from all three groups can identify if a certain agent/exposure can pose a carcinogenic risk; furthermore, it can gather mechanistic insights into the mode of the action of a specific carcinogen. This structured battery could be very useful for any type of in vitro study, containing human cell lines aiming to detect the potential carcinogenic risks of environmental agents/exposures. In fact, here, we include examples in which these approaches were successfully applied. Finally, we provide a series of advantages that, we believe, contribute to the suitability of our proposed approach for the evaluation of exposure-induced carcinogenic effects and for the development of an alternative strategy for conducting an exposure risk assessment.

## 1. Introduction

Chemical safety and exposure science have largely advanced by following a material-by-material approach and using conventional and robust assays established to assess well-known toxicity endpoints. However, the number of chemicals in the environment and in consumer products is overwhelmingly large, and some traditional assays are inefficient in evaluating the wide diversity of human exposures across individuals and throughout their lifetimes [1]. Most of these methodologies are animal-based, which are too resource-consuming to address all exposures and, further, present some limitations, such as the sometimes-conflictive extrapolation of results from the animal models to human populations and, most importantly, the ethical concerns that they present [2,3,4]. Moreover, many traditional approaches lack mechanistic insight and fail to mimic real-world exposure scenarios in which chemical mixtures and long-term/low-dose exposures are contemplated [5]. Thus, although the data gathered have great value, large knowledge gaps remain. As a result, the development of new approach methodologies (NAMs) is a field in need of expansion. NAMs comprise in silico, in chemico and in vitro approaches aiming to fill in those gaps in current exposure science to perform better-informed risk assessments and support regulatory decisions [6]. As defined by the ECHA (European Chemical Agency), NAMs include diverse and innovative high-throughput screening tools or high-content methods, such as genomics, proteomics and metabolomics, but also other more conventional methods able to inform on the toxicokinetic or toxicodynamics of substances in the context of hazard characterization [7]. The paradigm shift towards these new approaches is promoted by policies that call for the use of NAMs when suitable as a response to animal welfare issues, while also aiming to improve the speed and accuracy of the data necessary for safety evaluations. However, NAMs still exhibit several uncertainties and, thus, it is not currently possible to fully replace in vivo studies with alternative approaches [8]. Nonetheless, it is believed that applying an evaluation strategy that integrates data from several NAMs would efficiently allow us to prioritize data needed for the preliminary determination of the risks posed by exposures [9,10].

The development of NAMs becomes particularly important for the evaluation of the potential carcinogenic effects related to environmental exposures. Human populations are exposed daily to an ever-growing number of environmental pollutants. These environmental exposures are mainly characterized by two central aspects: (i) being long-term/low-dose exposures and (ii) being highly diverse [11]. Therefore, in the long term, there is a very likely scenario in which an agent—or a mixture—can initiate the carcinogenic process that develops over time. Up till now, in vivo studies have been the gold standard tool for carcinogenesis assessments, with rodents being the most popular model to evaluate the genotoxic and carcinogenic effects of exposures [4]. The two-year rodent bioassays are the accepted tests in standard OECD regulations to study the carcinogenic potential of chemicals [12]. In these assays, groups of animals are continuously exposed for most of their lifespan and are closely monitored to track any signs of toxicity or the development of neoplastic lesions. As previously mentioned, this approach presents important limitations: in vivo assays are too time-consuming, costly, and require many animals, which makes them impractical for the large testing programs desirable for carcinogenicity studies, despite efforts to implement the 3Rs principle (replacement, refinement and reduction). Therefore, approaches based on in vitro models are urgently required [13].

Among the in vitro alternative approaches to studying carcinogenicity, cell transformation assays (CTAs) are the most advanced in terms of standardization and validation. OECD (Organisation for Economic Co-operation and Development) test guidelines have been recommended for three CTAs able to identify agents with tumor-initiating and promoting activities [14,15,16]. They are:

*BALB/c3T3 CTA*: This assay predicts the tumor-initiating activity of an exposure. Briefly, following a standardized concentration range-finding experiment, A31-1-1 mouse fibroblasts are exposed for 72 h to the compound and are thereafter cultured in a fresh compound-free medium for 31 days. Then, the cells are fixed and foci are scored. Only the malignantly transformed BALB/c 3T3 cells form these foci, characterized by the multilayering of cells randomly orientated and morphologically divergent from the monolayer in the culture [17].

*Bhas 42 CTA*: This system was implemented to evaluate the tumor-initiating and promoting ability of the chemicals using the Bhas 42 cell line. This cell clone is prone to transformations, as it was isolated after transfecting BALB/c 3T3 mouse fibroblasts with an activated *ras* oncogene. Subsequent to the experiment carried out to find the optimal concentrations to be applied, the tumor-initiation capability of chemicals was determined by scoring the foci formed after an 18-day culture, following a 3-day exposure to the compound. In a modified version of this test, Bhas 42 cells were cultured for 3 days in a compound-free culture medium. Then, the medium was replaced with a compound-containing medium every 3 days until day 14, at which point the fresh compound-free medium was again added. On day 21, the cells were fixed and foci were scored to assess the tumor promotion ability of the exposure [18].

*SHE CTA*: The Syrian hamster embryo assay is based on primary cells isolated from Syrian hamster embryos, which form morphologically transformed colonies when exposed to an agent with carcinogenic potential. After choosing the concentration range, SHE cells were exposed to the test chemical from the third day of culturing and the exposure was continued until day 10. Then, the transformed cell colony frequency was quantified to determine the tumor-initiating potential of the exposures [19].

All the above-indicated tools are valuable as a first approach in carcinogenicity screening, contributing to a significant reduction in the number of agents that must undergo in vivo testing [15]. Nevertheless, CTAs do not completely mimic the whole carcinogenesis process as it happens in vivo and, in addition, they present important limitations [20], such as:

Performing single-endpoint analysis: Despite the multistep nature of carcinogenesis, these systems rely on the evaluation of a single endpoint related to cell transformation. Thus, information on other carcinogenic-related effects, mechanistic insights and potential differences in the transformed status of the exposed cells is missed.

Functioning with short-term periods of exposure: Although these approaches do not strictly belong to acute exposure systems, they far from mimic a real-life scenario, as only 3–10 days of exposure are contemplated and environmental exposures often last for long periods, even a lifetime.

Using animal cell models: The use of rodent cell clones can be very efficient in detecting rodent carcinogens; however, these systems can fail to identify specific human carcinogens [21]. Further, concerns regarding the extrapolation results to humans could arise as is the case for in vivo studies [22,23].

Despite the prevalidation efforts carried out by the European Center for Validation of Alternative Methods (ECVAM) [24], these limitations have hampered the extended use of CTAs. Overcoming these limitations by developing human-cell-line-based CTAs, for which different carcinogenic-related effects can be analyzed under a long-term exposure scenario, would greatly contribute to further improving the relevance of these in vitro approaches and increase the interest in incorporating CTAs into carcinogenicity testing strategies. Such efforts are ongoing as scientists aim to broaden the scope of CTAs by improving their prediction ability and including mechanistic explanations [25]. According to such expectations, this review aims to outline an in vitro long-term exposure approach and a battery of alternative CTAs that would fall under the umbrella of NAMs. In this scenario, not only the novel method of exposure is remarkable, but also the multistep endpoint analysis of the transformation biomarkers based on different hallmarks of carcinogenesis. Additionally, we include relevant studies in which these techniques were successfully used in the framework of exposure safety evaluations. Finally, we provide a series of advantages that show the suitability of our proposed approach for the evaluation of exposure-induced carcinogenic effects and for the development of an alternative strategy for assessing exposure risks.

## 2. Long-Term Exposures as a Novel Approach for Carcinogenesis Evaluation

Due to the progressive nature of cancer, the carcinogenic effects induced by environmental pollutants (chemicals or other agents) appear because of sustained exposures throughout extended periods of time. Accordingly, these types of exposures are especially relevant for in vitro scenarios, where the long-term accumulative effects under chronic exposure settings should be evaluated.

Long-term in vitro studies have successfully been used by different groups to determine the exposure-derived induction of cell transformation, the progression and the promotion of tumoral cells, and to establish potential mechanisms of action involved in the process at different time points (as discussed later). Following this approach, and as schematized in Figure 1, relevant target cells are chronically exposed to subtoxic concentrations of the tested agent for long periods of time, ranging from 6 to 30 weeks on average. Exposed cells are generally passaged weekly, and the agent-containing medium is changed once/twice a week to continuously maintain the selected exposure level, mimicking real-life scenarios. More importantly, nonexposed control cells must also be maintained in parallel for the selected number of weeks to discriminate the effects induced by the continuous passaging from those caused by the sustained exposure.

At different time points during the long-term in vitro studies, several hallmark biomarkers of the cell transformation process are evaluated (as developed later). This allows for the identification and description of the carcinogenic phenotype reached by the cells upon exposure over time, as well as the comparison of the transformed status at the initial, middle and end stages of the exposure period. More importantly, several authors have demonstrated that subsequent to prolonged exposure, the cells obtained in the in vitro cell transformation models induce tumorigenesis in mice [26,27,28,29], which is a conclusive marker of the identification of an agent able to induce a neoplastic transformation [30].

Long-term exposures reserve further uses. Thus, throughout the entire exposure time, cell samples can be collected to constitute a biobank for furthering the understanding of the mechanisms involved in the transformation process. Once the cell transformation occurs after long-term exposure to a specific agent, diverse tools, mainly molecular, can be used to identify the precise moment where different biomarkers started to be significantly altered. Moreover, this approach permits the use of cells maintained in the biobanks even after the completion of the study, until a time new tools become available to answer unsolved questions. In this way, DNA methylation changes have been associated with multiwalled carbon nanotube (MWCNT) long-term exposures [31], and alterations in the expression levels of specific genes (as *Mth1*) were observed in cells chronically exposed to ZnONPs and CoNPs [32], as well as changes in a panel of microRNAs being evaluated in cells previously long-term exposed to TiO_2_NPs and MWCNTs [33]. Further, the collection of samples for the application of omics analyses can greatly contribute to identifying key events and adverse outcome pathways, allowing to unravel the mechanisms of action.

## 3. Alternative Cell Transformation Assays Based on the Hallmarks of Carcinogenesis

The hallmarks of carcinogenesis have been deeply characterized after they were first defined [34] and extended [35]. Interestingly, a new paper on this topic has recently been published pointing out the hallmarks of cancer as an integrative concept to explain the complexity of the cancer process and to understand the mechanisms of cancer development and malignant progression more fully [36]. These hallmarks describe distinctive events of cancer development, namely, two enabling capabilities—(i) genomic instability and mutation and (ii) tumor-promoting inflammation—as well as eight hallmark capabilities—(i) sustaining proliferative signaling, (ii) enabling replicative immortality, (iii) evading growth suppressors, (iv) resisting cell death, (v) deregulating cellular energetics, (vi) avoiding immune destruction, (vii) inducing angiogenesis and (viii) activating invasion and metastasis. Mimicking the situation that takes place in vivo, the in vitro cell transformation is a process where cells acquire specific features that are representative of different stages during the progression of the tumoral phenotype and that differ from those of nontransformed cells. These phenotypic manifestations can be explored with different in vitro systems and, therefore, they are presented here as a battery of oncogenic biomarkers that provide information on the transformed status of the cells, as they align with different hallmarks of cancer.

Due to the progressive nature of the cell transformation, some of the proposed biomarkers are informative of the specific status of the cells during the acquisition of the tumoral phenotype. Accordingly, we grouped the different biomarkers in early, intermediate or advanced stages, which allowed for the identification of the cells in the initiation, promotion or aggressive stages of tumorigenesis, as schematized in Figure 2. Keeping this in mind, rather than focusing on single endpoints, the proposal was to use a battery of biomarkers under long-term exposure settings. In this way, the whole analysis would provide a more comprehensive view of the changes taking place during the carcinogenesis process. It is essential to select diverse endpoints from all categories of the battery of biomarkers to accurately characterize the in vitro transformation process at different time points to gather mechanistic insight and, ultimately, contribute robust information for risk assessment strategies.

### 3.1. In Vitro Cell Transformation Biomarkers and Methods

In the following sections, we described the rationale behind the different oncogenic biomarkers selected for our proposed battery. In addition, some of the available methodologies for their evaluation using in vitro approaches were also presented.

#### 3.1.1. Early Biomarkers

##### DNA Damage and Genotoxicity

Genetic instability is defined as a status of an increased propensity for genomic alterations, and it is one of the defining features of cancer. During the carcinogenic process, tumor cells accumulate both genetic and epigenetic alterations; thus, genetic and epigenetic instabilities are mechanisms intimately associated with carcinogenesis. DNA methylation processes prevent transcription factors from binding to DNA and, in addition, post-transcriptional histone modifications regulate the chromatin structure, having an important impact on biological processes such as gene expression, DNA repair and chromosomal condensation.

It was estimated that between 40 and 60 mutations occur in most solid tumors. However, not all occur at tumor initiation, but also later during the progression stage [37,38]. Therefore, assessing the genetic instability of cells during the in vitro transformation process is informative both at early and intermediate time points. Given that monitoring mutations in a nontargeted way is unrealistic, the evaluation of DNA damage and genotoxicity levels is an indirect biomarker of genetic instability that can eventually result in mutations [39]. Further, incorporating this endpoint into our proposed battery of assays allowed us to identify both genotoxic and nongenotoxic carcinogens, partly contributing to the mechanistic insight.

There are two gold standard methodologies for the assessment of DNA damage and genotoxicity: the comet assay and the micronucleus assay. OECD guidelines exist for both techniques [40,41], given their usefulness in detecting genotoxic agents and being surrogate biomarkers of carcinogenic risk. The comet assay is based on the migration of DNA strands of individual cells embedded in agarose. Briefly, the cells were fixed in agarose and placed on a support slide. Then, DNA was denaturized using alkaline lysis and a brief electrophoresis step was performed. If the DNA contained breaks, uncoiled DNA loops would migrate toward the anode, while migration would be prevented if the DNA was undamaged. Therefore, after fluorescent staining, the cells with damaged DNA would appear as “comets” with a bright head and a tail of variable intensities depending on the number of DNA breaks. From the initial protocol proposal [42], different modifications of the assay have been published, incorporating enzymes to detect oxidized DNA bases [43] or to improve the high throughput potential of the methodology and scoring [44,45].

On the other hand, the micronucleus assay measures the incidence of micronuclei (MNs), which are small chromatin bodies generated during cell division due to chromosomal fragmentation or chromosomal loss. Chromosomal fragmentation, more specifically double-strand breaks, leads to a structural chromosomal instability, characterized by ongoing errors in chromosomal segregation throughout successive cell divisions, providing a proliferative advantage to cancer cells. Consequently, this mechanism is particularly important in cancer progression.

Classically, MNs are manually scored through microscopy after inducing a cell cycle arrest [46]. Due to the labor-intensive nature of the MN analysis through the use of microscopy, alternatives based on the flow cytometry analysis have been proposed. This approach involves the lysis of outer membranes and the use of dyes to discriminate the MNs from the nuclei, based on the fluorescence intensity, which allows for a high-throughput analysis [47]. New variations of the MN assay are being proposed, aiming to further automatize the process and increment the high-throughput aspect of the technique [48].

##### Uncontrolled Proliferation

The proliferation of normal cell populations is controlled by a balance between growth-promoting and growth-suppressing signals, which allows for normal tissue homeostasis [49]. The dysregulation of these signals through diverse mechanisms leads to sustained cell growth, which is arguably the most identifying trait of cancer cells. Not for nothing, four out of the eight hallmark capabilities are related to uncontrolled cell proliferation, including sustained proliferative signaling, enabling replicative immortality, evading growth suppressors and resisting cell death [35]. Early in the tumorigenic process, cancer cells acquire the ability to constitutively activate proliferative signaling through autocrine signaling to their own growth factors, increased levels of mitogen receptors and the induction of ligand-independent or stroma signaling [50]. Moreover, transformed cells can evade cell cycle arrest signals and escape from cell death through the interaction of diverse signaling pathways at multiple levels, such as the overexpression of antiapoptotic molecules [51]. As a result, increased cell proliferation can be identified as an early cell transformation biomarker.

Diverse methodologies allow for the evaluation of cell proliferation, and many of them have been exhaustively reviewed [52,53]. The most common approach is the calculation of the population-doubling time, which estimates the time it takes for a cell population to double in size based on the number of cells in the population at two or more time points [54]. The population size can be estimated either through direct cell counting using a hemocytometer counting chamber or automated cell counters. However, aiming to move to high-throughput formats, quantitative methods compatible with multiwell plates have been developed. On the one hand, metabolic assays rely on the detection of a colorimetric signal proportional to the cells’ metabolic activity, allowing for the measurement of viable cells. MTT (3-(4,5-dimethylthiazol-2-yl)-2-5-diphenyltetrazolium bromide) [55], resazurin [56] or sulforhodamine [57] assays, among others, fall within this category of metabolic proliferation assays. Moreover, assays based on the DNA content and DNA synthesis are also direct and simple approaches to measuring cell proliferation. Within this category, the incorporation of BrdU (5-bromo-2′-deoxy-uridine) in DNA can be detected through the use of antibody-based techniques, in what is called the BrdU assay [58]. Additionally, in the EdU (5-ethynyl-2′-deoxyuridine) proliferation assay, EdU is incorporated into replicating DNA, providing a method to quantify and monitor proliferating cells through flow cytometry or microscopic approaches [59]. Further, in recent years, there has been an increment in the development of noninvasive technologies that allow for the real-time monitoring of cell proliferation in an incubator based on imaging [60] or measuring ultrasonic signals [61].

##### Morphological Changes

In culture populations, tracking the cell morphology is a very simple but informative biomarker of in vitro cell transformation. Early in the oncogenesis process, epithelial cells activate the epithelial-to-mesenchymal transition (EMT) program, which is characterized by major transcriptional and signaling alterations [62]. Eventually, these lead to phenotypical changes in epithelial cells, including the loss of polarity and morphological modifications; normal polygonal-shaped cells change, acquiring an elongated spindle-like morphology (Figure 3) [63].

In fibroblasts, morphological alterations indicate their activation into myofibroblasts, typically enriched in the tumor microenvironment [64]. Through the simple microscopic scoring of the cell population presenting an abnormal morphology, we could easily estimate the proportion of cells transitioning from a stationary state to becoming able to disseminate towards other tissues, or to promote the invasive potential of surrounding cells. The growing body of studies in the literature able to infer the metastatic potential of cells from diverse morphological parameters such as the angle, diameter or size supports the suitability of using cell morphologies as an in vitro transformation biomarker [65,66,67].

At an ultrastructural level, diverse changes in the mitochondrial structure have been observed as associated with tumoral cells [68,69]. Mitochondrial swelling is a characteristic feature of most cancer cells, but other alterations, such as the acquisition of an irregular shape or the disorganization and reduction in the cristae number, are also frequently reported. These morphological alterations have been correlated with mitochondrial dysfunction [70] and, therefore, are closely linked to a cancer hallmark capability, as it pertains to the deregulation of cellular energetics. Hence, we proposed the analysis of ultrastructural morphological changes as part of the battery of in vitro assays. Furthermore, the use of imaging techniques, such as transmission electron microscopy (TEM) or scanning electron microscopy (SEM), has also been long-used to evaluate ultrastructural alterations in very different cell types, and are still of great use [71,72].

##### Secretome Alterations

Tumor cells are not isolated, but rather subsist surrounded by a highly rich microenvironment of stromal cells formed by fibroblasts, pericytes, leukocytes, endothelial cells and other cell types [73]. The complexity of events that lead to the transformation of normal cells into tumoral cells is parallel to the evolution of the microenvironment into an activated state, triggered by the continuous crosstalk between the tumor and stromal cells [74]. This paracrine communication is based on the release of inflammatory cytokines (e.g., IL-6, IL-8 and IL-1β), growth factors, extracellular matrix (ECM)-remodeling enzymes and other soluble factors that constitute the cells’ secretome. During the process of oncogenic transformation, the secretome varies in composition, and can influence several hallmark abilities of tumoral cells, such as immune response modulation, angiogenesis, inflammation and invasiveness induction, therefore, influencing tumor promotion from the early stages of the transformation process [75]. Further, specific components of the secretome can actively drive cell transformation. This is especially important for the stem cell compartment, as regulation through soluble factors plays a major role in the cell fate determinant and in controlling their proliferation/differentiation balance. Bone morphogenetic proteins (BMPs) are representative of this role, given that they have been identified as key components of a permissive microenvironment that triggers stem cell transformation into cancer stem cells (CSCs) and allows for transformed cell persistence, as well as tumor progression [76].

Therefore, studying the microenvironment has become an important focus of interest in cancer research and, accordingly, we included it in the proposed battery of assays for the characterization of the oncogenic phenotype. The use of 3D (co)cultures is arguably the most suitable in vitro approach for determining the role of the microenvironment in tumor progression, as it allows for the stimulation of the tumor and stroma context by including ECM and different cell types in the model [77]. Nonetheless, here, we presented different alternatives for the study of the secretome in a 2D culture context, which is more adaptable for long-term exposure settings.

The conditioned media (CM) of starved cells is representative of the set of factors composing the secretome. Different approaches can be followed to analyze the CM and its potential influence on surrounding cells. The indirect soft-agar assay (Figure 4) allows for a global analysis of the secretome at the functional level. To proceed, the CM of the long-term exposed cells is mixed with the soft-agar substrate—a semisolid matrix containing agar and cell culture media [78]—to grow a model tumoral cell line (e.g., HCT-116 or HeLa cells) for 15 days. These cells are prone to colony formation on soft-agar; therefore, changes in colony number or size indicate an effect of the CM in promoting surrounding cell growth [79]. At a molecular level, changes in the levels of inflammatory mediators, such as IL-6, IL-10, TFG-β or TNF-α, among others, are indicative of an inflammatory microenvironment [80], which potentially contributes to cell transformation and tumor promotion, given the influence of chronic inflammation on cancer onset [81]. Therefore, using arrays for the evaluation of large sets of cytokine and soluble factors present in the CM is an informative approach (e.g., the human XL cytokine array kit from R&D Systems). The matrix metalloproteinases (MMPs) are ECM-remodeling enzymes sometimes represented in this kind of array. Increased MMP secretion and activity have been described in several types of cancers associated with poor prognosis, given their key role in ECM degradation, leading to angiogenesis and invasiveness promotion from the early stages of tumor progression [82,83]. The MMP activity in the cells’ CM can be specifically analyzed with substrate zymography. In that assay, proteins are run in an SDS-PAGE gel containing a specific substrate, namely, gelatin, casein or collagen. In this step, the proenzyme is activated, triggering the degradation of the substrate. Then, following a simple process of renaturation, development and staining, the areas of high protease activity can be identified as clear bands against a dark background [84]. Although MMP-2 and 9 are the most typically used markers, modifications in the zymography techniques have been implemented to increase the resolution for the analysis of other MMPs such as MMP-1, 8 or 13 [85].

#### 3.1.2. Intermediate Biomarkers

##### Anchorage-Independent Growth

In healthy tissues, cell-to-cell and cell–ECM adhesions are essential for maintaining homeostasis and development. However, during the transformation process, cells with an active EMT program adopt a more mesenchymal state by losing E-cadherin, increasing the expression of N-cadherin and altering the integrin expression profile [86]. Due to these changes, transformed cells acquire the ability to grow independently of anchorage. In epithelial tissue, this means that cells attach to the collagen in the ECM rather than to the basement membrane, contributing to ECM remodeling, which, eventually, facilitates tumoral cell dissemination [87]. This scenario can be reliably mimicked by the soft-agar assay, considered a stringent marker of in vitro malignant transformation. To perform this assay, cells are seeded in a semisolid matrix composed of two layers produced by mixing cell culture media and agar at different concentrations. The base layer has a higher agar concentration, while the top layer has less agar and contains the cells (Figure 5). After an incubation period of approximately 21 days, only cells with a transformed phenotype can grow and expand, forming colonies [78,88]. The usefulness of the soft-agar assay has been proved in multiple studies testing the potential carcinogenic effects of different agents, such as tobacco smoke components, where transforming biomarkers have been associated with several components of the gene signature, such as the AKT/mTOR signaling pathway that was dramatically activated [89]. Arsenic exposure, as a well-known human carcinogenic inductor, was also shown to induce alterations in the ability to grow independent of anchorage in different cell lines [79,90], and this ability grew in parallel with the accumulation of oxidatively damaged DNA [91].

##### Migration Potential

In an intermediate stage of malignant transformation, cells that undergo EMT activation, morphological changes and the loss of substrate attachment initiate the invasion–metastasis cascade. This involves a succession of events beginning with a local invasion at the tumor site, followed by extravasation into distant tissues and the eventual formation of metastatic lesions that can generate secondary tumors [92]. Therefore, in the progressive activation of the cell metastatic potential, two distinct processes can be identified: (i) cell migration, defined as the ability of cells to move on a substrate, such as the intact basal membrane or the ECM, and (ii) cell invasion, which involves the degradation of the substrate infiltrated by the cells [93].

In vitro-transformed cells also progressively acquire metastatic potential. The migration ability of cells can be evaluated with previously reviewed diverse approaches [94,95]. One of the most used methodologies to assess the cell migration potential is the wound-healing assay or scratch assay, consisting of creating a scratch in the confluent cell monolayer and periodically capturing images to monitor cell movement into the scratched area [96,97,98]. Different modifications of this protocol have been proposed to increase the reproducibility of the assay, such as using silicone inserts to increment the consistency of the scratch [99] and using automated imaging systems [100] or video microscopy [101], which ensure the continuous monitoring of cell migration in identical sections. Another typically used approach is the Boyden chamber assay or transwell migration assay, where transwell inserts are placed in the culture wells to create a chamber with an apical part separated from the basolateral compartment by a porous membrane. Cells are generally seeded on the apical side in a starvation medium, while a medium containing a chemoattractant or a higher serum concentration is added to the basal part (Figure 6). Therefore, cells with an active migration potential translocate vertically through the porous membrane. To determine the migration rate, cells attached to the basal side of the membrane and those that detached and reached the basal compartment should be quantified, for which colorimetric or cell-counting methods can be applied [97]. Using these approaches, the migration rate of cells can be quantified at a low cost and with an easy setup and readout, although plenty of alternative methodologies exist, such as the impedance-based real-time migration measurement [102] or single-cell tracking [103].

#### 3.1.3. Advanced Biomarkers

##### Invasion Potential

As previously mentioned, during the transformation process, cells progressively acquire metastatic potential. One of the advanced stages of the invasion–metastasis cascade and of the carcinogenic process is the acquisition of invasion potential. Invading cells can penetrate tissue barriers by degrading basal membranes and the ECM, which allows for the tumor cell infiltration of secondary tissues [93]. Therefore, the invasion potential can be considered a biomarker of an advanced and aggressive phenotype.

Cell invasion can be evaluated following similar in vitro approaches to those selected for the migration analysis, but incorporating some kind of ECM-like substrate that the cells must degrade to move through the surface. To apply the Boyden chamber invasion assay, the setup is identical to the transwell migration assay, with the exception that a layer of a synthetic matrix mimicking the ECM is added on top of the porous membrane (Figure 7). Only invasive cells degrade the ECM layer and translocate through the membrane, while other cells are blocked from migrating [97,104,105]. The impedance-based real-time migration measurement [102] or single-cell tracking [103] can also be adapted to measure cell invasion by adding an ECM coating on the surface. Further, for those cells able to form spheroids, several variations of spheroid invasion protocols can be found in the literature. Globally, to follow this approach, cells are grown as multicellular spheroids and embedded into ECM gels. After 72–96 h of incubation, invading cells or cell clusters spread out on the spheroid bodies, while noninvasive cells remain, maintaining the compact spheroid structure [106,107]. It is important to point out that this kind of 3D model better mimics physiological tissue structures than 2D approaches, as cell-to-cell contacts are stronger and there is an irregular distribution of oxygen and nutrients. Moreover, different cell types can be incorporated into the spheroids, aiming to reproduce a more realistic tumor microenvironment and analyze the stromal/tumor cell interaction during cell invasion [108].

##### Stem-like Features

Accumulating evidence supports the idea that cancer stem cells (CSCs) are a normal constituent of many tumors, contributing to their heterogeneity. CSCs are a subpopulation of cells that present or acquire stem-cell-like properties, capable of self-renewal, but also divide asymmetrically, giving rise to the bulk of the tumor [109]. In the tumor microenvironment, both nontumoral and differentiated tumoral cells interact with CSCs, modulating key oncogenic events, such as the tumor growth, metastatic potential, treatment resistance and tumor recurrence [110].

During the in vitro carcinogenesis process, tumoral stem-like cells can be enriched and identified with the tumorsphere formation assay. To reach such an aim, a low density of cells is seeded in ultra-low attachment plates under specific culture conditions achieved by adding several growth factors to the serum-free media. Under such conditions, tumor spheres are formed [111,112]. These are solid, spherical structures easily distinguishable from cell aggregates, because no individual cells can be identified (Figure 8). The number of tumorspheres formed can be used to estimate the proportion of stem-like cells present in a population of transformed cells. Tumorspheres derived in vitro from tumoral cells have been described to be highly invasive and to initiate tumorigenesis when engrafted in vivo [113]. Thus, the acquisition of this sphere-forming capability is considered a marker of aggressiveness and CSC enrichment in the in vitro cell-transformed population.

As a summary of the previous sections, we included a table (Table 1) showing the different described biomarkers of in vitro transformation, the proposed assays and the references containing the protocols explaining how the different assays can be carried out.

## 4. Representative Studies Using the Proposed Approach to Evaluate In Vitro Carcinogenesis

In the literature, we could find several studies that successfully characterized the tumoral phenotype of cells by applying the in vitro long-term exposure approach together with the analysis of oncogenic biomarkers. In this section, we presented representative works developed in the context of the carcinogenic risk evaluation. The selected studies applied a low range of exposures lasting for at least 6 weeks on different in vitro systems to draw a conclusion on the cell-transforming potential of certain environmental exposures based (at least partially) on two or more of the assays previously presented here.

Several of the chosen studies focused on chemicals. Interestingly, the use of in vitro systems to study carcinogenesis has allowed for characterizing the carcinogenic potential of classical environmental pollutants such as arsenic. Thus, a study carried out [115] showed that 26 weeks of low-concentration chronic arsenic exposure increased the proliferation and anchorage-independent growth of target pulmonary cells (A549) and suggested that cell transformation was mediated by increased levels of intracellular reactive oxygen species (ROS) and related signaling. The involvement of oxidative stress in the arsenic-induced transformation was evaluated by Bach et al. [79], who showed that oxidative DNA damage-sensitive mouse embryonic fibroblasts (MEF *Ogg1^−/−^*) were more prone to transformation than their wild-type counterparts (MEFs *Ogg1^+/+^*) after 40 weeks of subtoxic arsenic exposure. MEF *Ogg1^−/−^* cells acquired a clear spindle-like morphology and increased the proliferation, while this effect was not observed in the exposed MEFs *Ogg1^+/+^*. Although both cell types acquired anchorage-independent growth and tumor-promoting abilities through the increased secretion of MMP2 + 9, DNA damage-sensitive cells were more prone to the transformation being positive for the biomarkers analyzed 10 weeks earlier during the exposure period. That study highlighted the relevance of oxidative damage in arsenic-induced carcinogenesis and, further, allowed for the creation of a cell biobank with the established cell model, which was, subsequently, used to expand the mechanistic explanation by describing the role of *FRA1* [32] and *MTH1* [33] in the process. Additionally, in this context, the work by Weinmuellner et al. [116] reported that after 24 weeks of chronic arsenic trioxide (ATO) exposure, immortalized human keratinocytes acquired a spindle-like morphology, migration potential and an enhanced sphere-forming capacity. The acquisition of these oncogenic traits was observed in parallel to an increase in the ROS levels, which, again, was evidence of arsenic-induced oxidative stress. Further, the development of oncogenic features was confirmed by observing that the transformed cells were able to form tumors in SCID mice. However, tumor formation in the SCID mice was transient, which brought the authors to conclude that chronic exposure to low-level arsenic concentrations led to the development of a mild malignant phenotype that may require other factors (e.g., UV radiation or mechanic stress) for a full-blown malignant transformation. Interestingly, the in vitro long-term exposure approach was also applied to identify the chemopreventive ability of black tea extract (BTE) in a BTE and arsenic coexposure scenario [117]. In that study, skin keratinocyte cells (HaCaTs) exposed to arsenic for 34 weeks showed clear signs of transformation, while cells coexposed to arsenic and BTE for the same period did not develop such evident malignant features. Among the hallmarks altered by arsenic exposure, the authors described accelerated proliferation; modified cell morphology associated with the loss of contact inhibition; an altered expression of EMT markers (Vimentin, Snail, Slug, Twist, Zeb and N-cadherin); elevated MMP2 + 9 expression and activity, explored with Western blot and zymography, respectively; increased migration potential observed both through the wound-healing and transwell-based approaches; and high invasion potential evidenced by using the transwell invasion assay. However, all these transformation signs were ameliorated in the BTE + arsenic exposure conditions, which indicated the potential chemopreventive role of BTE in arsenic-induced carcinogenesis.

The carcinogenic potential of other chemicals has also been evaluated following the long-term/low-concentration exposure approach. Here, we highlighted some studies that focused on chemicals with very diverse applications. As a first example, aluminum salts (AlCl_3_) are frequently present in industrial products of daily use, including food additives and cosmetics. In vitro studies using murine and human mammary cell models have shown that long-term exposure to AlCl_3_ induces alterations in hallmark traits in the cells. Mandriota et al. [28] showed that normal murine mammary gland epithelial cells suffered morphological changes and acquired the ability to grow independent of anchorage in the soft-agar assay, as well as the ability to form large metastatic tumors in immunodeficient mice. In the same direction, Sappino et al. [118] described that AlCl_3_ induced double-strand breaks and increased proliferation, the loss of contact inhibition and anchorage-independent growth of human mammary epithelial cells (MCF-10A). The use of mammary cells was previously found to be useful by Zou and Matsumura [119], who used MCF-7 cells to compare the effects of exposure to the pesticide contaminant β-hexachlorocyclohexane (β-HCH) with those induced by estrogen 17b-estradiol (E_2_), which was used as a positive control due to its protumorigenic effects in breast tissue. The authors evaluated the cellular phenotype at different time points. After 7 months of exposure, only E_2_ induced the alteration of oncogenic biomarkers. Nonetheless, the exposure to both E_2_ and β-HCH for 143 months triggered important phenotypical changes in the cells, including enhanced levels and activity of MMP-9 and increased anchorage-independent growth and invasiveness. Taken together, these studies evidenced that selecting relevant target models and time points allows to identify the potential detrimental effects that challenge the safety of widespread chemicals, in these cases regarding the pathogenesis and progression of breast cancer. Other interesting studies have focused on the long-term in vitro effects of tobacco components. As an example, two components of second-hand smoke, namely, nicotine (Nic) and 4-(methylnitrosamino)-1-(3-pyridyl)-1-butanone (NNK), were shown to have an additive effect compared with the carcinogenicity of the compounds separately. Fararjeh and colleagues [120] exposed human breast epithelial cells (HBL-100) to very low concentrations of Nic and NNK alone or in combination for 9 weeks. The results showed that tumor progression features emerged earlier in coexposed cells, which also acquired a more marked tumoral phenotype with significant increases in cell proliferation, anchorage-independent growth, migration and stem-cell-like properties. Similarly, the compound 4-aminobiphenyl (4-ABP), which is another component of tobacco smoke, as well as of cooking oil and dyes, was found to induce liver carcinogenesis in an HBx Src (p53^−/−^) transgenic zebrafish model after four months of exposure; nevertheless, when wild-type zebrafish were used, the exposure required seven months to show its tumoral potential [121]. Interestingly, when the authors carried out an in vitro study using hepatic cell lines (like HepG2 and L-02), significant increases in cell proliferation, the number of colonies formed in the soft-agar assay and the number of migrating cells in the transwell migration assay were observed. These effects were reported after exposures lasting for 8 weeks to 10 nM 4-ABP. The enhancement of the oncogenic biomarkers was observed in a concentration-independent manner likely due to the stronger toxicity of 4-ABP concentrations above 10 nM. As another example, when L-02 liver cells were exposed for up to 20 passages to musk xylene, which is a common component of personal care products, high levels of cell proliferation, altered cell morphology and an increased anchorage-independent growth were observed. It is interesting to remark that when the clones obtained in the soft-agar assay were expanded, they showed very significant migration and invasion potentials. The authors also reported that these in vitro transformation effects were found in parallel to the repression of the TGF-β pathway [122]. It is of note that the application of the long-term exposure approach has also proven useful in identifying the impact of chemicals on the cells’ response to alterations in their surrounding microenvironment. Clément et al. [123] exposed a model of mammary stem cells (MCF10A) to very low concentrations of bisphenol A (BPA) or benzo(a)pyrene (B(a)P) for 8.5 weeks. The results showed that the exposure primed the cells’ response to BMPs, which, as previously mentioned, are key regulators of stem cells and their niche, and play a role in the initiation of stem cell transformation [124,125]. While BPA and B(a)P did not significantly change the phenotype of the cells, the long-term exposure enhanced the protumoral effects induced by high levels of BMPs, including a deregulated proliferation and tumorsphere formation ability [123]. Therefore, that work suggested that chemical-induced alterations both in the stem cells and their microenvironment could synergistically drive cell transformation.

The use of in vitro models for a carcinogenic assessment has also contributed greatly to the field of nanoparticle safety evaluation. Nanoparticles (NPs) are becoming increasingly present in consumer products and medical applications due to their advantageous physicochemical properties. The diversity and complexity of these materials call for the development of novel methods with high-throughput potentials, such as the approach presented here. We were able to find several examples in the literature where a long-term/low-concentration approach was applied to evaluate the transforming potential of different NPs. Thus, Annangi and coworkers used the DNA damage-sensitive cells introduced previously (MEF *Ogg1*^−/−^) to be chronically exposed to cobalt (CoNPs), zinc oxide (ZnONPs) and ZnCl_2_ (as the ionic form of zinc). After 12 weeks of exposure, CoNPs induced the malignant transformation of cells, changing their morphology, increasing MMP2 + 9 secretion and promoting colony formation in the soft-agar assay [126]. However, when cells were exposed to both ZnONPs and ZnCl_2_ under the same exposure scenario, no increases in the selected oncogenic biomarkers were observed. These negative effects were observed despite the toxic potential found upon acute exposure [127]. On the contrary, ZnONPs were reported to induce the malignant transformation of colonic mucosal cells (IMCEs), which showed hyperproliferation, morphological changes, an anchorage-independent growth ability in the soft-agar assay and migration potential (as shown with the wound-healing assay). In addition to the different biomarkers analyzed in the in vitro carcinogenesis study, the tumoral phenotype of the cells after 30 passages of exposure was confirmed, given their tumorigenic potential when subcutaneously administered to nude mice [128]. These conflicting results highlighted the importance of cell model selection, given that different cell types may present different levels of sensitivity to a specific exposure.

Interestingly, lung cell models are often used in the nanotoxicology field, since inhalation is considered one of the main routes of NP exposure. Thus, bronchial epithelial cells (BEAS-2B) have previously been selected as models in several studies reporting on the in vitro carcinogenic potential of diverse NPs. Wang and collaborators [27] performed a comprehensive study of the carcinogenic potential of single-walled carbon nanotubes (SWCNTs), which have a resemblance to asbestos. In this work, BEAS-2B cells were chronically exposed to a low concentration of SWCNTs for 6 months. The long-term exposure triggered the acceleration of cell doubling, induced morphological changes, increased the anchorage-independent growth in the soft-agar assay, enhanced the migration and invasion potential in transwell-based assays and, interestingly, promoted angiogenic activity. Once the cells had achieved a malignant in vitro phenotype, they were subcutaneously injected into nude mice, resulting in the formation of tumors that did not develop in the control mice. Further, the authors explored p53 signaling, which has been suggested as a potential mechanism driving the carcinogenic effect of SWCNTs. Regarding other NPs generally deemed as safe, it was found that 6 weeks of exposure to CeO_2_NPs, cigarette smoke condensate (CSC) and the combination of both resulted in an altered cell transformation status of BEAS-2B cells in a concentration-dependent manner. More importantly, the coexposure induced an enhanced malignant phenotype, as shown by the assessment of early, intermediate [129] and advanced in vitro transformation biomarkers [130]. Similarly, the same cell model was selected by Gliga et al. [131] to evaluate the transforming effects of NiNPs and their derivatives (NiONPs and NiCl_2_, as the ionic form) after exposures lasting for 6 weeks. Although the exposure induced significant DNA damage and gene expression changes, only a tendency to increase the number of colonies and the proportion of invading cells was identified with the soft-agar and the transwell invasion assay, respectively. A more recent study reported on the transforming potential of NiNPs in the same cell line, but upon a longer period of exposure. Just two biomarkers of in vitro cell transformation were analyzed after 21 weeks of exposure, and both were positive, namely, DNA damage and anchorage-independent growth. Nonetheless, a defect in the DNA repair (HIF-1α/miR-210/Rad52 pathway) was identified as a contributing mechanism of Ni-induced carcinogenesis and confirmed with knock-out approaches (of the hypoxia-inducible factor 1 (HIF) and *mir-210*), which significantly reduced the cell transformation effects [132]. Lastly, BEAS-2B cells have also been used to evaluate the protective effects of nanoparticle coatings. The work by Kornberg et al. [133] compared the transforming effects of a specific type of iron oxide NPs (Fe_2_O_3_NPs), the same particle, but coated with amorphous silica (SiO_2_-Fe_2_O_3_NPs), and gas metal arc welding fumes generated from mild steel welding (GMA-MS), which contained insoluble iron. After 28 weeks of low-concentration exposure to Fe_2_O_3_NPs, the exposed cells presented a similar phenotype to that induced by GMA-MS. This was characterized by accelerated cell proliferation, the induction of DNA damage and anchorage-independent colony formation. These biomarkers were not significantly altered in BEAS-2B cells exposed to SiO_2_-Fe_2_O_3_NPs, which highlighted the importance of the physicochemical properties of NPs in their associated risks.

Regarding the potential of the in vitro carcinogenesis tools as a first approach to fill in current knowledge gaps, of note is their usefulness in determining potential health effects associated with exposures to emergent relevant pollutants, such as micro/nanoplastics (MNPLs). The field of MNPL risk assessments is currently in expansion, although most of the reported studies focus on short-term endpoints. Few publications using long-term exposures were found in the literature, and they mostly focused specifically on oxidative stress induction and the genotoxic aspect of the exposure [134,135,136]. Nonetheless, the carcinogenic potential of polystyrene nanoplastics (PSNPLs) has been explored using a sensitive pretransformed cell model called prone-to-transformation progress (PTP) cells, derived from the previously mentioned DNA damage-sensitive MEFs. This cell line is especially sensitive to oxidative damage induction. The results obtained using this cell line showed that a 12-week coexposure to PSNPLs and arsenic enhanced the malignant features of the phenotype, significantly increasing the proportion of spindle-like cells, the levels of DNA damage, the colony-formation ability in soft-agar and the percentage of migrating and invading cells. Interestingly, PSNPLs and arsenic alone did not induce the promotion of intermediate and advanced transformation biomarkers [137]. However, PSNPL exposure for an extended period (up to 24 weeks) did induce several cancer hallmarks, such as anchorage-independent cell growth, promoted cell migration and invasion ability, as well as an increase in the stem-cell-like population. Furthermore, several micro-RNAs, stress-related genes and pluripotency markers were also abnormally expressed in the same cell type [114].

Table 2 summarizes the main characteristics of the studies reported in this section.

## 5. Conclusions and Further Steps

Our proposal solved the old topic of how to use in vitro approaches to determine the potential carcinogenic risk of environmental agents/exposures. Due to the complexity of the tumoral process, trying to identify it by using a simple endpoint (foci formation) seems to be a bit naïve; thus, the proposed/prevalidated CTAs have not yet received the general approval of the scientific community. Consequently, few studies have been carried out/published using these approaches. This is a serious issue, given that everybody agrees that detecting the potential genotoxic and/or carcinogenic risk of environmental agents, including those of emergent concern, is an urgent task. In this context, we consider that our proposal of using a structured battery of assays, detecting different biomarkers representative of known cancer hallmarks, is undoubtedly useful in covering the existing gaps. This proposal could pose a framework inspiring those researchers involved in hazard assessments, mainly focusing on cancer risk assessments. It is obvious that an in vitro approach does not completely mimic all the changes that occur in a complex in vivo model, as in mammals. Nevertheless, as a screening tool, our proposal could identify agents with carcinogenic potential, as well as their potential mechanisms of action. According to the reported data, further in vivo approaches could be planned if required.

In addition to the well-recognized mechanisms underlying all the proposed biomarkers of in vitro transformation, our proposal focused on the need of using long-term approaches to determine such effects. In a good experimental approach, extending exposures for several weeks, ideally between 15 and 30 weeks, are necessary. It is important to maintain constant exposure conditions and concurrent controls to distinguish if the observed effects were due to the exposure or to the long-term growth conditions. Moreover, it would be very useful to retain a biobank of cells at different time points (e.g., every 5 weeks) to identify the evolution of the different biomarkers or, ideally, to be used in the future when new and more sophisticated tools are available. To better simulate a real exposure scenario, long-term exposures should be associated with low-concentration (nontoxic) exposures. Thus, the low concentrations and the long-term exposures are a binomial that should preside over these types of experiments looking for the in vitro carcinogenicity potential of environmental agents/exposures. The selection of these concentrations should consider the cytotoxicity induced by the agent in the cell system of choice, and, if available, the data on the agent’s environmental and/or internal exposure levels. Thus, the selected range of concentrations should set a maximum limit at the concentration able to cause cytotoxicity, and should be coherent with a realistic human exposure scenario. 

Another advantage of our approach was the possibility to desist the use of rodent cells, such as those used in the prevalidated CTAs. For our proposal, any type of human cell can be selected, although generally aiming to use a cell line representative of the main target of the evaluated agent, if known (e.g., lungs for inhalation exposures). As ingestion and inhalation are the most important exposure routes, cells representative of such tissues can be selected as the first exposure targets. Nevertheless, if the final target (e.g., the kidneys) is known, representative cells from such organs can also be used. This flexibility permits the design of experimental approaches far from the corset of using only one type of cell line. Independent of the selected target cells, an interesting point to be highlighted is that to increase the sensitivity of the assays, cell lines deficient in specific pathways associated with the expected mechanisms of action of the evaluated agents could be very useful. In fact, this approach was used in the prevalidated CTAs. Thus, for example, since the induction of oxidative stress is a mechanism associated with the mode of the action of different carcinogens, the use of cell lines deficient in their ability to repair oxidative damage could be a good option. In addition, using models of human stem cells is of major importance for long-term effect assessments, as this rare subset of cells is the only one to persist in the body for many years, allowing abnormalities to be accumulated. In the field of cancer, this implies that stem cells are major targets from which transformed cells emerge. Outside this specific disease, stem cells are gatekeepers of tissue homeostasis, and perturbing their function likely impairs many aspects of human physiology. These are only examples of the many possibilities that this kind of in vitro approach provides. Following this approach, researchers can explore multiple options, including different agents or mixtures, concentrations, times of exposure, endpoints and mechanisms of action. All of this would ideally fill in urgent knowledge gaps in exposure science and contribute to the emergence of an alternative strategy for conducting exposure risk assessments.

## Figures and Tables

**Figure 1 ijms-24-07851-f001:**
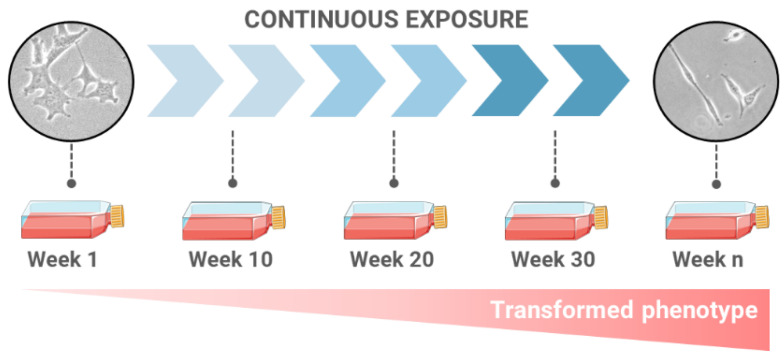
In vitro long-term/low-concentration studies. Cells progressively develop a transformed phenotype when cultured for long periods of time whilst chronically exposed to low concentrations of a carcinogenic agent.

**Figure 2 ijms-24-07851-f002:**
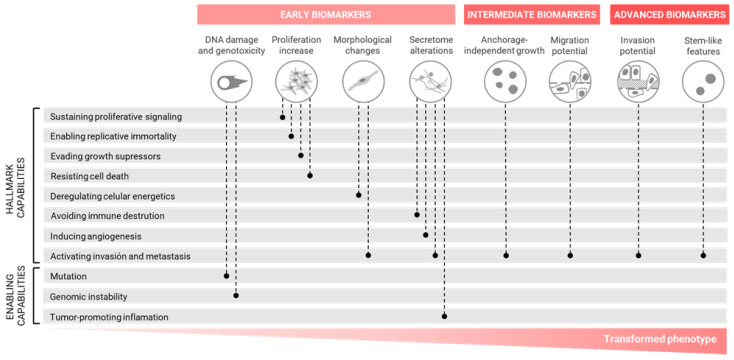
A battery of oncogenic biomarkers. The proposed biomarkers were grouped into early, intermediate and advanced stages, as they were representative of different stages during the progression of the transformed phenotype. Each biomarker is associated with one or more hallmarks of cancer and enabling capabilities.

**Figure 3 ijms-24-07851-f003:**
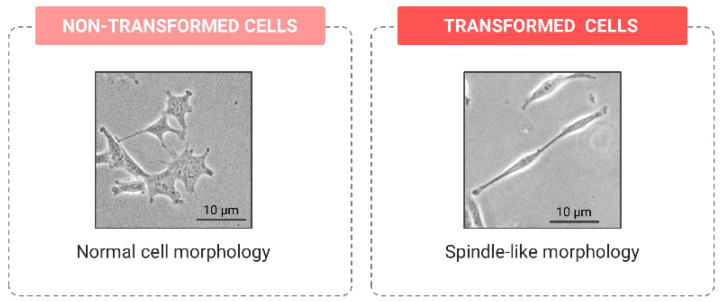
Comparison between the cell morphology of non-transformed cells and the spindle-like shape of transformed cells.

**Figure 4 ijms-24-07851-f004:**
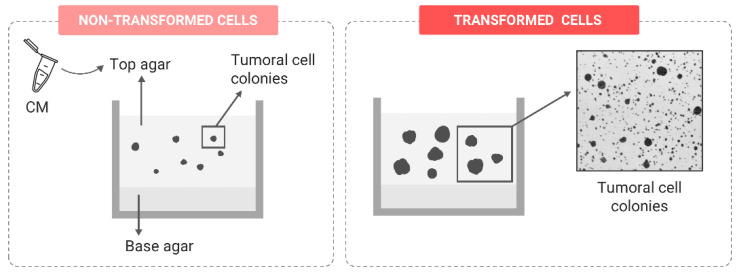
Growth of tumoral cells (Hela or HCT116) in the indirect soft-agar assay in the presence of CM collected from the culture of non-transformed or transformed cells.

**Figure 5 ijms-24-07851-f005:**
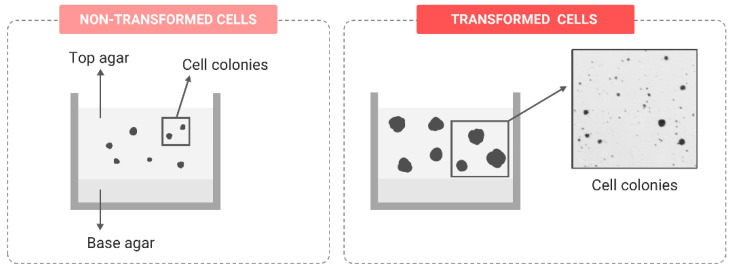
Representation of cell colony growth in the direct soft-agar assay comparing the result before and after transformation.

**Figure 6 ijms-24-07851-f006:**
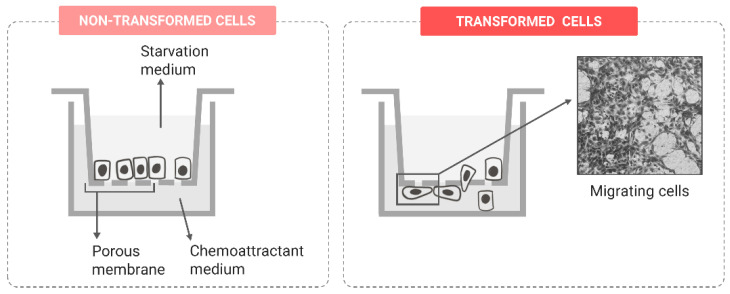
Comparison of the Boyden chamber migration assay in transformed and non-transformed cells.

**Figure 7 ijms-24-07851-f007:**
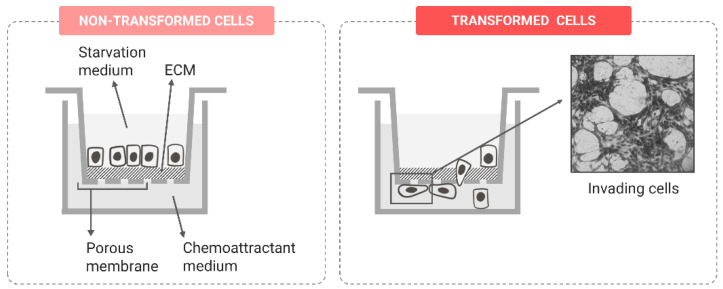
Representation of the Boyden chamber invasion assay with transformed and non-transformed cells.

**Figure 8 ijms-24-07851-f008:**
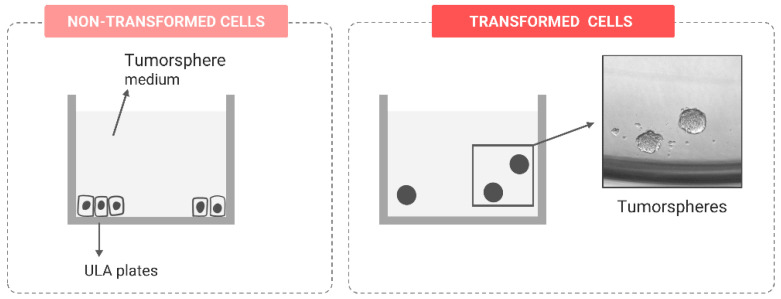
Tumorsphere formation assay in ultra-low attachment plates (ULA). Only transformed cells form tumorspheres.

**Table 1 ijms-24-07851-t001:** Methodologies for the evaluation of the oncogenic biomarkers.

Biomarkers of In Vitro Transformation		Assays	Protocols
Early biomarkers	DNA damage and genotoxicity	Comet assay	[42,43,44,45]
		Micronucleus assay	[46,47,48]
	Uncontrolled proliferation	Population-doubling time	[54]
		Metabolic assays	[55,56,57]
		DNA synthesis and quantification assays	[58,59]
		Real-time monitoring of cell proliferation	[60,61]
	Morphological changes	Microscopic imaging	[65,66,67]
		Ultrastructural imaging (TEM and SEM)	[71,72]
	Secretome alterations	Indirect soft-agar assay	[78,79]
		Cytokine arrays	[114]
		Zymogram	[84]
Intermediate biomarkers	Anchorage-independent growth	Soft-agar assay	[78,88]
	Migration potential	Wound-healing assay	[96,97,98,99,100,101]
		Boyden chamber assay	[97]
		Impedance-based real-time migration assay	[102]
		Single-cell tracking	[103]
Advanced biomarkers	Invasion potential	Boyden chamber assay	[97,104,105]
		Impedance-based real-time migration assay	[102]
		Single-cell tracking	[103]
		Spheroid invasion assay	[106,107,108]
	Stem-like feature acquisition	Tumorsphere formation assay	[111,112]

**Table 2 ijms-24-07851-t002:** Studies that applied the in vitro carcinogenesis approach for safety evaluation.

Reference	Agent	Cell Line	Time of Exposure	Concentration	Early Biomarkers				Intermediate Biomarkers		Advanced Biomarkers		Transformed Phenotype
					DNA Damage and Genotoxicity	Uncontrolled Proliferation	Morphological Changes	Secretome Alterations	Anchorage-Independent Growth	Migration Potential	Invasion Potential	Sem-Like Features Acquisition	
[126]	CoNPs	DNA damage-sensitive MEFs	12 weeks (3 months)	0.05 μg/mL			+	+	+				Yes
				0.1 μg/mL			+	+	+				Yes
[127]	ZnONPs	DNA damage-sensitive MEFs	12 weeks (3 months)	1 μg/mL	-		-	-	-				No
	ZnCl_2_				-		-	-	-				No
[79]	As	MEFs	40 weeks (9 months)	0.5–2 μM		-	-	+	+				Mild
		DNA damage-sensitive MEFs				+	+	+	+				Yes
[130]	CeO_2_NPs	BEAS-2B	6 weeks (1.5 months)	2.5 μg/mL CeO_2_ 5 μg/mL CSC							+	+	Yes
	CSC										+	+	Yes
	CeO_2_NPs + CSC										+	+	Yes
[137]	PSNPLs	Prone-to-transformation progress MEFs	12 weeks (3 months)	25 μg/mL PSNPLs 2 μM As	+	-	+		-	-	-	-	Mild
	As				+	-	+		-	-	-	-	Mild
	PSNPLs + As				+	-	+		+	+	+	-	Enhanced
[114]	PSNPLs	Prone-to-transformation progress MEFs	24 weeks (6 months)	25 μg/mL					+	+	+	+	Yes
[115]	As	BEAS-2B	26 weeks (6.5 months)	0.25 μM		+			+				Yes
				1 μM		+			+				Yes
				5 μM		+			+				Yes
[123]	BPA	MCF-10A	8.5 weeks (2 months)	10^−10^ M BPA 10^−10^ M B(a)P		+*						+	Enhanced
	B(a)P					+*						+	Enhanced
[120]	Nic	HBL-100	9 weeks (2 months)	1 mM Nic 100 femtM NNK		-			+	+	-	+	Yes
	NKK					-			+	+	+	+	Yes
	Nic + NKK					+			+	+	+	+	Enhanced
[117]	As	HaCaT	34 weeks (8 months)	100 nM As 1 μM BTE		+	+	+		+	+		Yes
	As + BTE					-	-	-		-	-		Prevented
[131]	NiNPs	BEAS-2B	6 weeks (1.5 months)	0.5 μg/mL	+	-			+ *	-	+ *		Mild
	NiONPs				+	-			+ *	-	+ *		Mild
	NiCl_2_				+	-			+ *	-	+ *		Mild
[133]	Fe_2_O_3_NPs	BEAS-2B	28 weeks (6.5 months)	2.88 μg/mL	+	+			+				Yes
	SiO_2_- Fe_2_O_3_NPs				-	-			-				No
	GMA-MS				+	+			+				Yes
[127]	4-ABP	L-02	8 weeks (2 months)	10 nM		+			+	+			Yes
		HepG2				+			+	+			Yes
[126]	AlCl_3_	NMuMG epithelial cells	16 weeks (4 months)	100 μM		-	+		+				Yes
[128]	ZnONPs	IMCE	30 passages	1 μg/mL		+		+	+	+			Yes
[132]	NiNPs	BEAS-2B	21 weeks (5 months)	0.25 μg/mL	+				+				Yes
				0.5 μg/mL	+				+				Yes
[129]	CeO_2_NPs	BEAS-2B	6 weeks (1.5 months)	2.5 μg/mL CeO_2_ 5 μg/mL CSC		-	-	-	-	-			No
	CSC					-	+	+ **	+	+			Yes
	CeO_2_NPs + CSC					+	+	+	+	+			Yes
[118]	AlCl_3_	MCF-10A	9 weeks (2 months)	10–300 μM	+	+	+		+				Yes
[27]	SWCNT	BEAS-2B	24 weeks (6 months)	0.02 μg/cm^2^		+	+		+	+	+		Yes
[116]	ATO	Immortalized human keratinocytes	24 weeks (6 months)	0.05 μM			+			-		+	Mild
				0.1 μM			+			+		+	Mild
				0.25 μM			+			+		+	Mild
[122]	Musk xylene	L-02	20 passages	10 μg/L		+	+		+	+	+		Yes
				100 μg/L		+	+		+	+	+		Yes
				1000 μg/L		+	+		+	+	+		Yes
[119]	E_2_	MCF-7	56 weeks (13 months)	1 nM E_2_				+	+		+		Yes
	β-HCH			100 nM B-HCH				+	+		+		Yes
				1 μM B-HCH				+	+		+		Yes

+: significantly increased result for the respective biomarker; -: unchanged result for the respective biomarker; *: no statistically significant result, but a clear tendency was described; **: disparity in the results of two different assays performed to evaluate the same oncogenic biomarker.

## Data Availability

Data will be available on request.

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
