# Peer review of "In Vitro Approaches to Determine the Potential Carcinogenic Risk of Environmental Pollutants"

_ijms, 2023, doi:10.3390/ijms24097851_

Round 1

Reviewer 1 Report

I believe that in the section on DNA damage and genotoxicity, that the authors add a paragraph on potential epigenetic alterations, according to the new findings on DNA modifications.

Regarding DNA fragmentation, the authors must include a specific paragraph of results for cancer cells.

I consider that in the reflection of the conclusions they are important. Especially the implications in relation to renal cells, and / or lung.

Minimal corrections to the English language must be made.

Author Response

See the attached document

Reviewer 2 Report

In the paper, authors discuss in detail about the In vitro Approaches to Determine the Potential Carcinogenic 2 Risk of Environmental Pollutants. Overall manuscript is well written some of the minor grammatical corrections are required to polish the manuscript. 

Author Response

See the attached document

Reviewer 3 Report

It is very important to use a suitable method to assess potential carcinogen toxicity risk of environmental exposed pollutants.  In this manuscript (ijms-2324482), the authors reviewed the current methods and proposed a general battery containing assay which can be potentially used for different types of in vitro studies. This is a very intereting idea, which would enrich the area of the in vitro study. And the authors also list the advantages of this method. Does this method has some potential problems or drawbacks? If the authors could list some potential or possible drawbacks of the method  in the manuscript, that will be much helpful.

Author Response

See the attached document
